# Comparing the Fasting and Random-Fed Metabolome Response to an Oral Glucose Tolerance Test in Children and Adolescents: Implications of Sex, Obesity, and Insulin Resistance

**DOI:** 10.3390/nu13103365

**Published:** 2021-09-25

**Authors:** Jennifer L. LaBarre, Emily Hirschfeld, Tanu Soni, Maureen Kachman, Janis Wigginton, William Duren, Johanna Y. Fleischman, Alla Karnovsky, Charles F. Burant, Joyce M. Lee

**Affiliations:** 1Department of Internal Medicine, University of Michigan Medical School, Ann Arbor, MI 48109, USA; burantc@med.umich.edu; 2Department of Medicine, Dartmouth-Hitchcock Medical Center, Weight and Wellness Center, Lebanon, NH 03766, USA; 3Susan B Meister Child Health Evaluation and Research Center, Department of Pediatrics, University of Michigan, Ann Arbor, MI 48109, USA; ehirschf@med.umich.edu; 4Michigan Regional Comprehensive Metabolomics Resource Core, University of Michigan, Ann Arbor, MI 48109, USA; tanusoni@umich.edu (T.S.); mkachman@umich.edu (M.K.); wiggie@med.umich.edu (J.W.); wld@med.umich.edu (W.D.); 5Department of Computational Medicine and Bioinformatics, University of Michigan Medical School, Ann Arbor, MI 48109, USA; akarnovs@med.umich.edu; 6Department of Molecular and Integrative Physiology, University of Michigan, Ann Arbor, MI 48109, USA; johf@umich.edu; 7Division of Pediatric Endocrinology, Department of Pediatrics, University of Michigan, Ann Arbor, MI 48109, USA

**Keywords:** oral glucose tolerance test, metabolomics, fatty acids, adolescents, acylcarnitines, obesity, insulin resistance, glucose challenge

## Abstract

As the incidence of obesity and type 2 diabetes (T2D) is occurring at a younger age, studying adolescent nutrient metabolism can provide insights on the development of T2D. Metabolic challenges, including an oral glucose tolerance test (OGTT) can assess the effects of perturbations in nutrient metabolism. Here, we present alterations in the global metabolome in response to an OGTT, classifying the influence of obesity and insulin resistance (IR) in adolescents that arrived at the clinic fasted and in a random-fed state. Participants were recruited as lean (*n* = 55, aged 8–17 years, BMI percentile 5–85%) and overweight and obese (OVOB, *n* = 228, aged 8–17 years, BMI percentile ≥ 85%). Untargeted metabolomics profiled 246 annotated metabolites in plasma at t0 and t60 min during the OGTT. Our results suggest that obesity and IR influence the switch from fatty acid (FA) to glucose oxidation in response to the OGTT. Obesity was associated with a blunted decline of acylcarnitines and fatty acid oxidation intermediates. In females, metabolites from the Fasted and Random-Fed OGTT were associated with HOMA-IR, including diacylglycerols, leucine/isoleucine, acylcarnitines, and phosphocholines. Our results indicate that at an early age, obesity and IR may influence the metabolome dynamics in response to a glucose challenge.

## 1. Introduction

As the prevalence of prediabetes and type 2 diabetes (T2D) in adolescents is increasing [1], it is vital to identify metabolic dysfunction prior to disease onset to classify individual risk and implement preventative strategies. Classically, an oral glucose tolerance test (OGTT) diagnoses impaired glucose tolerance (IGT)/prediabetes and T2D, by measuring the acute trajectory of glucose in response to ingestion a 75-g glucose solution. Profiling the metabolome in response to an OGTT can provide a deeper phenotyping of T2D risk, expanding upon measuring traditional glucose levels to a profile of metabolic response across multiple pathways. Several studies have demonstrated the widespread response of the metabolome to an OGTT in adults, observing alterations in proteolysis, lipolysis, ketogenesis, and glycolysis in healthy individuals in response to the challenge [2,3]. These studies suggest an acute increase of glycolytic intermediates and rapid inhibition of lipolysis [4] and proteolysis [5], as evidenced by decreases in amino acids, free fatty acids (FFA), and acylcarnitine (AC) intermediates of beta-oxidation. 

Adolescent obesity is a strong risk factor for the development of T2D, as a nationwide study observed that severe obesity increases the incidence of T2D in early adulthood in both males and females [6]. In fasted plasma, obesity is associated with alterations in the metabolome, including elevations in lipids, branched chain amino acids (BCAA), and aromatic amino acids [7,8,9]. In adults, the metabolome response to an OGTT was profiled in the context of obesity, observing a delayed reduction of FFA and higher levels of amino acids, including isoleucine and leucine, at 30 min post-OGTT in adults with obesity [10]. In adolescents, Müllner et al. [11] identified metabolites in response to an OGTT associated with obesity, including elevations of AC 2:0, glutamate, alanine, and pyruvate, suggesting a mismatch between beta-oxidation and TCA-cycle activity. 

Pediatricians are more likely to order non-fasting tests compared with a gold standard fasting OGTT because of the inconvenience of fasting tests [12]. As a result, a random plasma glucose sample and a 1-h non-fasting glucose challenge have been assessed for the prediction of T2D, offering strong discrimination for identifying prediabetes in adolescents [13]. Furthermore, evidence has suggested a tight link between post-meal glucose levels and T2D complications [14], supporting the utility of a random-fed glucose challenge. Although the fasted and non-fasted metabolome have established differences [15], it is uncertain how the metabolome response to an OGTT differs in individuals arriving to the clinic fasted and in a random-fed state. In addition, the influence of insulin resistance (IR) on the fasted vs. random-fed metabolome response to an OGTT is unknown. At fasting, multiple metabolic pathways are associated with IR and T2D including BCAA metabolism [16], beta oxidation [17], and bile acids [18]. In adults arriving to the clinic fasted, Nowak et al. observed correlations between AC response to the OGTT with a degree of IR, as individuals with worsening IR had a blunted decline of medium-chain ACs, products of beta-oxidation [19]. 

Our main objectives were to assess the metabolome response to an OGTT by (1) comparing the response to a fasted OGTT in participants with overweight and obesity with that in controls and (2) determining differential metabolite responses to a fasted vs. random-fed OGTT among participants who were overweight and had obesity. Furthermore, we sought to identify metabolite responses to an OGTT associated with insulin resistance (as measured by HOMA-IR) in participants who were overweight and had obesity at the Fasted and Random-Fed Visit, considering the influence of sex. Results from these analyses identified a panel of metabolites that can be profiled by fasted or random glucose challenges with the potential to predict longitudinal T2D risk. 

## 2. Materials and Methods

### 2.1. Research Design

The cohort consisted of adolescents who were either overweight or obese (OVOB) (BMI percentile ≥85th for sex/age [20]) and lean adolescents (BMI percentile <85th for sex/age [20]), aged 8–17 years at time of enrollment, recruited from primary care and pediatric specialty clinics in southeast Michigan (2015–2018). Individuals were excluded if they had known diabetes, use of medications known to affect glucose metabolism (oral steroids, metformin, insulin, or sulfonylureas), verbal report of pregnancy, or acute or chronic infections. Written informed consent was obtained from the parent/guardian for all participants and participants ≥10 years provided written assent. The study was approved by the University of Michigan Institutional Review Board.

Participants attended study visits at the Michigan Clinical Research Unit, where a medical history, vital signs, anthropometrics, and laboratory evaluation were performed. During the Fasted OGTT Challenge, OVOB (*n* = 228) and lean participants (*n* = 55) arrived after an overnight fast for a formal OGTT (Figure 1), with fasting times ranging from 9 h and 35 min to 19 h and 21 min (Appendix A). The OGTT dosage consisted of 1.75 g glucose/kg body weight, with the maximum dosage of 75 g glucose (Glucola, Fisherbrand) (Appendix A). Blood samples were drawn at baseline (t0) and every 30 min (t30, t60, t90, and t120 min) following the challenge. Glucose and insulin levels were profiled from blood samples collected at t0, t30, t60, t90, and t120 min following the OGTT. The untargeted metabolome was profiled from blood samples collected at t0 and t60 min following the fasted OGTT. Approximately a week later, OVOB individuals (*n* = 228) returned for a glucose challenge in a random fed state, where participants were not given instructions on the timing of their last meal (50-g, Random-Fed OGTT challenge) (Figure 1), with fasting times varying from 5 min to 14 h and 16 min (*n* = 166 reported last mealtimes) (Appendix A). Our objective was to mimic a random OGTT that is already performed in the clinic as a screening test for gestational diabetes. The 50-g random OGTT has been previously shown have reasonable discrimination for identifying children with prediabetes [13]. Glucose and insulin levels were profiled from blood samples collected at t0 and t60 min following the OGTT. The untargeted metabolome was profiled from blood samples collected at t0 and t60 min following the Random-Fed OGTT. 

### 2.2. Laboratory Measurements 

The Michigan Diabetes Research Center (MDRC, Ann Arbor, USA) laboratory performed glucose homeostasis assays. Glucose was measured using the glucose hexokinase method and run on a Randox rX Daytona chemistry analyzer (Randox Laboratories Limited, Crumlin, UK). Insulin was profiled using a double-antibody radioimmunoassay [21]. The homeostatic model assessment for insulin resistance (HOMA-IR) utilized glucose and insulin measurement to estimate insulin resistance and beta cell function [22]. Glucose area under the curve (AUC) was estimated between t0 and t120 min by integrating, using the trapezoid method and stopping when glucose values dropped below baseline (GraphPad PrismVersion 8.4.3). Insulin AUC was estimated between t0 and t120 by integrating using the trapezoid method (GraphPad PrismVersion 8.4.3). Hemoglobin A1c (HbA1c) was determined using a Tosoh G7 HPLC Analyzer (Tosoh Biosciences Inc., San Francisco, CA, USA). Biologically implausible outliers were removed (*n* = 1 removed from t0 insulin value). Impaired fasting glucose (IFG) was defined as fasting plasma glucose ≥ 100 mg/dL; impaired glucose tolerance (IGT) was defined as the 2-h glucose ≥ 140 mg/dL; and prediabetes was defined as IFG, IGT, or HbA1c between 5.7–6.4% [23].

### 2.3. Untargeted Metabolomics 

Untargeted metabolomics analyses were performed by the Michigan Regional Comprehensive Metabolomics Resource Core (MRC^2^) (Ann Arbor, MI, USA). Metabolites were extracted from plasma samples using a solvent of methanol, acetonitrile, and acetone (1:1:1) including internal standards (100 mL extraction solvent and 4 mL internal standards). Samples were reconstituted with a solvent containing methanol and H_2_0 (2:98). Untargeted metabolomics was performed on an Agilent system consisting of an Infinity Lab II UPLC coupled with a 6545 qTOF mass spectrometer (Agilent Technologies, Santa Clara, CA) using a JetStream electrospray ionization source. The eluent was analyzed in both positive and negative ion mode electrospray ionization. Chromatographic peaks, representative of metabolite features, were detected using a modified version of existing commercial software (Agilent MassHunter Qualitative Analysis). Data normalization accounted for drift removal within and between batches by utilizing pooled reference samples that were analyzed within each batch using the Systematic Error Removal using Random Forest (SERRF) method. Metabolites were identified via comparing their MS/MS spectra to authentic standards, purchased internal or external standards ran on the same instrument. For this analysis, annotated metabolites (*n* = 246) were selected. Missing peak intensities were imputed by K-nearest neighbor (K = 5) in metabolites with ≥70% detection across samples. R package “impute” was used for imputation. Metabolites with less than 70% detection across samples were removed. Biologically implausible metabolite peak intensity values were removed. 

### 2.4. Statistical Analyses 

Descriptive statistics were computed for categorical variables (Pearson’s chi-square test) and continuous variables (unpaired Students’ *t*-test), stratified by OVOB and lean. Sex-stratified analyses were evaluated. Main analysis objectives are outlined in Appendix A. Peak intensities (PI) of metabolites were utilized for statistical analyses. 

Metabolite differences were identified between OVOB and lean at the Fasted Visit (Appendix A). At t0, linear regression models were run assessing the effect of group (ß_group_, OVOB or lean) on metabolite levels (log2 and standardized), adjusting for sex and age at the Fasted Visit.
(1)$$Metabolite={ß}_{o}+{ß}_{group}X+{ß}_{age}X+{ß}_{sex}X+\epsilon_{i}$$

Differential metabolites were identified using an adjusted *p*-value (false discovery rate [FDR] < 0.1) [24]. Positive ß_group_ values represent elevations in OVOB and negative ß_group_ values represent lower in OVOB. Differential metabolites were selected for metabolite set enrichment analysis (MSEA) [25] to identify biologically meaningful pathways associated with BMI in the metabolomics data. Human Metabolome Database (HMDB) IDs were mapped to 58 of the 66 differential metabolites at t0. Pathway enrichment analysis used the Small Molecular Pathway Database (SMPDB), which includes 99 metabolite sets based on normal human metabolic pathways. Over Representation Analyses (ORA) with the hypergeometric test was used to determine if metabolite pathways are represented more than expected by chance, denoting significance using a one-tailed *p*-value (unadjusted and FDR reported). Enrichment analyses were run through Metaboanalyst 4.0 [26]. At t0, sex differences in the metabolome were considered using unpaired Student’s *t*-test (Appendix A). Differential metabolites were identified using an adjusted *p*-value (FDR < 0.05).

Paired *t*-tests distinguished metabolites that significantly differed between t0 and t60 in each group and state (lean-Fasted, OVOB-Fasted, and OVOB-Random-Fed) using untransformed PI (FDR < 0.05) (Appendix A). Fold changes were calculated to represent metabolite response using log2(t60 PI/t0 PI). To assess if the metabolite response to the OGTT was associated with BMI group, unpaired *t*-tests distinguished variations in metabolite fold changes between OVOB and lean at the Fasted Visit (FDR < 0.1) (Appendix A).

OVOB participants returned to the clinic for a Random-Fasted OGTT Challenge. Differences in glucose and insulin levels between the Fasted and Random-Fed OGTT Challenge were computed (paired *t*-test). Linear regression models were run separately at t0 and t60 assessing the influence of state (ß_state_, Fasted or Random-Fed) on metabolite levels (log2 and standardized across state at each time), adjusting for sex and age at the Fasted Visit (Appendix A),
(2)$$Metabolite={ß}_{o}+{ß}_{state}X+{ß}_{age}X+{ß}_{sex}X+\epsilon_{i}$$

Differential metabolites were identified using an adjusted p-value (FDR < 0.1). Positive ß_state_ values represent elevated in OVOB-Fasted and negative ß_state_ values represent elevated in OVOB-Random-Fed. 

In OVOB individuals, metabolites were identified that are associated with IR, measured by HOMA-IR (Appendix A). Sex stratified models were run considering differences in glucose homeostasis measures by sex. Linear regression models were run separately at the Fasted Visit (t0, t60, fold change) and the Random-Fed Visit (t0, t60, fold change) on metabolite levels (log2 and standardized across each time and state), adjusting for age at the Fasted Visit.
(3)$$HOMA-IR={ß}_{o}+{ß}_{metabolite}X+{ß}_{age}X+\epsilon_{i}\;$$

Differential metabolites were identified using an adjusted *p*-value (FDR < 0.1). All statistical analyses were performed in R version 4.0. 

## 3. Results

### 3.1. Participant Characteristics 

Descriptive characteristics of the study population from the Fasted OGTT Challenge are reported in Table 1. Most of the cohort was Caucasian and non-Hispanic and included more females (*n* = 160) than males (*n* = 123), with similar distributions of sex, race, and ethnicity between OVOB and lean groups. No group trend was observed between IGT, IFG, and prediabetes status, with 15% of OVOB and 12% of lean having prediabetes. Three of the six lean participants who classified as prediabetic had a BMI percentile of 84%, potentially explaining why group trends were not observed. No group differences were observed in glucose levels during the OGTT (Figure 2) or glucose response measured by AUC glucose. Group differences in the insulin response to the OGTT were observed, with a larger insulin response, measured by AUC insulin, and higher insulin levels beginning at t0 and continuing through the completion of blood draws in OVOB (Figure 2). Sex differences were observed with higher levels of glucose t120 (*p* = 0.024), insulin t90 (*p* = 0.002), insulin t120 (*p* < 0.001), and BMI percentile (*p* = 0.007) in females and higher levels of glucose t30 (*p* = 0.003) in males (Appendix A). Females had a larger insulin response to the OGTT than males, measured by insulin AUC (*p* = 0.016) (Appendix A), suggesting a small decrease in insulin sensitivity. 

### 3.2. Influence of Obesity and Sex on the Fasting Metabolome

At the Fasted Visit, 66 metabolites were significantly associated with BMI group (OVOB vs. lean) at t0, adjusting for sex and age (Appendix A). Select differential metabolites by BMI group are shown in Figure 3. Short-chain ACs were elevated in OVOB compared to lean, including AC 3:0, 5:0, and 5:0-OH, representing alterations in BCAA metabolism. No differences were observed in BCAAs, potentially because our analysis did not account for muscle mass differences between OVOB and lean [7]. The aromatic amino acid tryptophan and its metabolite kynurenine was significantly elevated in OVOB, in contrast to the literature that consistently shows phenylalanine and tyrosine being elevated with obesity [27]. The biomarker 3-indolepropionate, a tryptophan metabolite that has been associated with a reduced likelihood of developing T2D [28], was significantly elevated in lean participants. Very long-chain FAs were elevated in lean at t0, with no BMI group differences observed in long-chain FAs. In lean participants, there were higher levels of beta-oxidation AC intermediates (AC 10:0, AC12:0, AC 14:1, AC 14:2, AC 16:0, and AC 18:2) and omega oxidation dicarboxylic FAs (FA 10:0-COOH, FA 11:0-COOH, and FA 16:0-COOH), potentially suggesting increased flux through FA oxidation pathways in lean individuals at fasting. Several lysophospholipids were elevated in lean compared to OVOB at t0, including lysophosphocholine (LPC) 16:0, LPC 17:0, LPC 18:1, LPC 18:2, LPC 20:0, and LPE 18:2, which parallels studies in pediatrics [8] and adults [29]. Multiple lipid species, including diglycerides (DG), phosphocholine (PC), and sphingomyelin (SM), were elevated in OVOB individuals, due to elevation in fat mass and consistent with previous observations [7]. Chenodeoxycholate (CDCA), a primary bile acid synthesized in the liver, was elevated in OVOB at t0. Multiple conjugated bile acids were differential between BMI groups at t0, including glycocholate, taurocholate, and tauro-alpha/beta-muricholate were elevated in lean and hyodeoxycholate was elevated in OVOB. 

Metabolites significantly associated with BMI group at t0 (FDR < 0.1) were selected for metabolite set enrichment analysis (MSEA) to identify biological pathways enriched with obesity (Appendix A). No pathways reached an adjusted significance threshold, although Beta Oxidation of Very Long Chain Fatty Acids was trending towards significantly enriched at t0 (unadjusted *p* = 0.07). 

As sex is associated with BMI percentile and glucose homeostasis measures within this cohort (Appendix A), metabolites were identified that were differential by sex at t0 during the Fasted Visit. Using unpaired *t*-tests, 40 metabolites were associated with sex (FDR < 0.05), with higher levels of FA, SM, and PCs in females and higher levels of short-chain ACs and amino acid metabolites (e.g., kynurenine and 3-methyl-2-oxovalerate) in males (Appendix A). These associations emphasize variations in fat and muscle mass in adolescents during puberty, as previous findings detail differential metabolites elevated in obese males and females [7]. Sex-stratification will be considered in additional analyses. 

### 3.3. Metabolome Response to the OGTT in OVOB and Lean Participants

The response of the metabolome to an OGTT is represented in Figure 4, stratifying individuals by state (Fasted and Random-Fed), time (t0 and t60), and group (lean and OVOB). Metabolites peak intensities were centered across all samples and metabolites were grouped using hierarchical clustering to identify groups of metabolites with similar dynamics during the OGTT. Significant changes in metabolite levels from t0 to t60 are reported, including alterations in 68% of metabolites in lean during the Fasted OGTT Challenge (Appendix A), 84% of metabolites in OVOB during the Fasted OGTT Challenge (Appendix A), and 77% of metabolites in OVOB during the Random-Fed OGTT Challenge (Appendix A) (FDR < 0.05). Most metabolites decreased in response to the OGTT, which may be attributed to the high abundance of lipids within the annotated metabolites in this dataset. Metabolite classes that consistently decreased in response to the OGTT include medium- and long-chain ACs, FFA, and lipids, such as SMs, PCs, and DGs. The metabolite with the largest increase was hippurate, increasing at approximately 4 log2FC in each group. As reported by Shaham et al. [2], this likely reflects the metabolism of the preservative benzoic acid, found in the glucola beverage used for the OGTT [30]. All paired *t*-tests are reported in Appendix A.

Using a fold change, differences in metabolite response between BMI groups at the Fasted Visit was explored, identifying 15 significant metabolites (Appendix A). Five medium- and long-chain ACs (AC 8:0, 10:1, 12:1, 14:2, and 16:1), two fatty acid oxidation intermediates (FA 10:0-COOH and FA 16:0-COOH), and six FAs (FA 12:0, 12:2, 14:2, 20:0, 20:1, and 22:1) had a larger decrease in lean than OVOB. These results suggest a more robust decrease in lipolysis and beta-oxidation in lean in response to the glucose challenge. 

### 3.4. Metabolome Differences between the Fasted and Random-Fed OGTT Challenge in OVOB

OVOB participants returned to the clinic for a random-fed state OGTT challenge. At t0, random-fed OVOB participants had significantly higher glucose (*p* = 0.0052) and insulin levels (*p* = 3.28^−^^20^) than individuals in the fasted state (Appendix A). At t60, random-fed participants had significantly lower levels of glucose (*p* = 1.40^−21^) than individuals in the Fasted OGTT Challenge, although their insulin levels were not significantly different (*p* = 0.657). These results suggest individuals arriving to the clinic for the OGTT in a variety of fed states have a primed insulin response, enabling the rapid response to the glucose load. Metabolites were identified that were associated with arriving to the OGTT Challenge fasted or random-fed at t0 and t60. At t0, 155 metabolites (63% profiled) and at t60, 122 metabolites (49% profiled) differed between OVOB-Fasted and OVOB-Random-Fed (Appendix A). Grouping by super pathway, metabolites are represented indicating direction of association (ß_state_) and significance (−log10 [*p*-value]) for t0 (Figure 5A) and t60 (Figure 5B). 

At t0, almost all the FAs profiled (97%) were significantly higher at the Fasted Visit compared to the Random-Fed Visit, as expected, representing mobilization of energy substances (anabolism) from adipose tissue during fasting. In parallel, additional lipids were higher at the Fasted Visit, including all SMs and 94% of the profiled PCs (15 PCs). Lysophospholipids varied in their associations with state, with five higher in the fasted state (LPC 16:0, LPC 17:0, LPC 18:1, LPC 20:0, LPC 23:0) and two higher in the random state (LPC 18:2 and LPE 18:2). Most medium- and long-chain ACs were higher at fasting, paralleling the FA levels and representing increases in beta-oxidation at fasting [31]. Interestingly, three of the four dicarboxylic fatty acids profiled were higher at the Random-Fed state at (FA 9:0-COOH, FA 10:0-COOH, and FA 11:0-COOH), suggesting increased omega oxidation related to the fed state, perhaps due to increased carbohydrate oxidation reducing the capacity to oxidize FA still entering the system. Many amino acids were higher at the Random-Fed Visit, including histidine, isoleucine/leucine, methionine, proline, tryptophan, tyrosine, and valine. Several short-chain ACs were higher in at the Random-Fed Visit, indicating increased BCAA metabolism. Bile acids, including primary, secondary, and conjugate bile acids, were higher at the Random-Fed Visit, representing bile acid and gut hormone response to a meal [32]. 

At t60, the metabolome represents the switch from an anabolic to a catabolic state in response to the glucose challenge. A portion of the differential metabolites at t0 normalized between OVOB-Fasted and OVOB-Random-Fed, including most of the differential lipids (SMs and PCs). At t60, 75% of the FAs and 78% of the medium- and long-chain ACs remain higher at the Fasted Visit. Fatty acid oxidation intermediates varied in their association with state at t60, with several higher in the Fasted group (FA 9:0-COOH, FA 12:0-OH, FA 12:0-NH2, and FA 14:0-OH) and several higher in the Random-Fed group (FA 10:0-OH and FA 10:0-COOH). Seven of the twelve lysophospholipids profiled were higher at the Random-Fed Visit at t60. Amino acids and bile acids remained higher in Random-Fed at t60.

### 3.5. Sex-Specific Associations of Metabolite Trajectories with Insulin Resistance in Participants with Overweight and Obesity

In OVOB, metabolites were identified from the fasted (t0, t60, and fold change) and random-fed (t0, t60, and fold change) glucose challenges that were associated with IR, measured by HOMA-IR (Appendix A). Sex stratified linear models were used considering the differential glucose and insulin responses between males and females (Appendix A). All results are reported in Appendix A. 

In males across all visits and time points, no metabolites were significantly associated with HOMA-IR. The metabolite mesobilirubinogen was trending towards positive association with HOMA-IR at t60 in the Fasted and Random-Fed visit (FDR < 0.2). 

In females, metabolites within multiple pathways were correlated with HOMA-IR at t0 and t60 during the Fasted and Random-Fed visits (Table 2). Consistently, diacylglycerides (DG 32:0, DG 32:1, DG 34:1, and DG 34:2) and the nucleotide urate were positively associated with HOMA-IR, the latter supported by previous work establishing the connection between hyperuricemia and IR [33]. More specifically, at t60 during the fasted visit, several amino acid metabolites (isoleucine/leucine, AC 5:0-OH, proline, and glutamate) and lipids (DGs and PCs) were positively associated with HOMA-IR. These results expand upon previous work [3], which found a blunted decrease in levels of BCAAs and other amino acid metabolites in subjects with IR. Represented by the fold change from the fasted visit, medium- and long-chain ACs were positively associated with HOMA-IR, demonstrating that a blunted decline in ACs in response to a glucose challenge is associated with IR, which parallels the decline in FA following the glucose challenge. Comparing the significant metabolites at Fasted t60 vs. the fold change during the fasted visit, only DG 32:0 and PC 32:1 were significantly associated with HOMA-IR in both models. Although less metabolites were significantly associated with HOMA-IR during the Random-Fed Visit, at t60, DGs, monoglycerides, glutamate, and urate exhibited positive associations. No significant associations were observed using the fold change from the Random-Fed.

## 4. Discussion

In the present study, we have characterized the metabolome response during an OGTT in OVOB (*n* = 228) and lean adolescents (*n* = 55). We identified metabolites that change significantly during the glucose challenge, highlighting the switch from FA to glucose oxidation at 60 min during the OGTT. We classified differential metabolites by BMI status at baseline and during the OGTT, suggesting that at an early age, obesity and its metabolic consequences may influence the metabolome dynamics in response to a challenge. Subsequently, overweight and obese adolescents returned to the clinic for a random-fed glucose challenge to compare the fasted and random-fed metabolome to degree of IR, and significant associations were found in female participants but not in males. Our results are the first study to deeply assess the fasted and random-fed metabolome response in adolescents and will be used for future analyses predicting the longitudinal risk of prediabetes development within the cohort.

### 4.1. Lipids, Fatty Acids, and Acylcarnitines

In response to the glucose challenge, most lipids, FAs, and FA oxidation intermediates, including hydroxyl-FAs, dicarboxylic FAs, and acylcarnitines, decreased. As observed in previous studies in adults, these alterations in the metabolome are reflective of the switch from FA oxidation to glucose oxidation and fat storage during the OGTT [34]. Acylcarnitines, biomarkers of mitochondrial beta-oxidation, reflecting the relative utilization of FA to carbohydrate [34] and reflect the degree of IR [35]. At the Fasted-Visit, several medium- and long-chain ACs and dicarboxylic FAs were lower in OVOB participants (Figure 3) and had a blunted decline in OVOB participants (Appendix A). Furthermore, in OVOB females at the fasted visit, the fold change of eight ACs (AC 5:1, 6:0, 12:0, 12:1, 14:0, 16:0, 16:1, and 18:0) was positively associated with HOMA-IR (FDR < 0.1). These results suggest that starting at a young age, obesity and IR influences metabolic flexibility in response to a glucose load [36]. In parallel, Nowak et al. [19] in a group of older males observed that AC 10:0 and AC 12:0 exhibited a smaller decline at 30 min in response to an OGTT, suggesting that the sustained elevation of the AC may directly impair insulin sensitivity. Our findings suggest that during adolescence, the prolonged insulin response (Figure 2) in OVOB females is also associated with insulin resistance. 

At the fasted visit at baseline, lipids, including DGs and SMs, exhibited positive associations with obesity (Appendix A), supported by previous analyses [7]. A primary question in these studies is whether the non-fasted state could be used to identify changes in metabolism in relation to IR. In females, at both the Fasted and Random-Fed Visit at t0 and t60, diacylglycerides (DG 32:0, DG 32:1, DG 34:1, and DG 34:2) were positively associated with HOMA-IR, suggesting that independent of fed-state, these lipids may provide predictive ability for the progression of IR and T2D. 

### 4.2. Amino Acids

In lean adolescents, approximately half of the profiled amino acids and their metabolites decreased in response to the OGTT, including leucine/isoleucine, methionine, histidine, serine, and glutamate, representing a decrease in proteolysis [2,3]. Deviations in amino acids response were observed between OVOB and lean, potentially due to the elevated insulin response within OVOB (Table 1, Figure 2). The larger insulin response in OVOB may act on skeletal muscles to decrease protein degradation [37], as evidenced by significant decreases in amino acids, including trypthophan, lysine, and glutamine, in only OVOB adolescents at t60. Comparing the metabolome response to an OGTT in 14 obese and 6 lean adults, Geidenstam et al. [10] observed at 30 min post-OGTT that asparagine, glutamate, taurine, tyrosine, and leucine/isoleucine increased in obese adults, which was absent in lean. This effect was not evident in our cohort, potentially because the metabolome was profiled at a later timepoint (t60). 

At t60 during the fasted visit, several amino acids (leucine/isoleucine, glutamate, and proline) and amino acid metabolites (gamma-glutamyltyrosine, L-gamma-glutamylisoleucine and N-acetylphenylalanine) were associated with IR in females (Table 2). Without stratifying by sex, Mullner et al. observed levels of BCAAs associated with a heightened insulin response [11]. Frequent inconsistencies in the association between BCAA and IR in adolescents are observed [7,38], due to study population differences in age, sexual maturation, and degree of IR, representing major challenges in paediatric prediction studies. Glutamate was associated with HOMA-IR in the Fasted and Random-Fed OGTT at t0 and t60. In a rate-limiting TCA cycle step, alpha-ketoglutarate is converted to glutamate-by-glutamate dehydrogenase, allowing for a rescue pathway of excess TCA substrate. Elevated levels of glutamate have been associated with an increased risk of T2D [39], as our results highlight the sensitivity between IR and TCA cycle overload in females. Overall, the t60 metabolome at the Fasted Visit had the largest number of significant amino acids with IR in females, suggesting a lack of suppression of proteolysis with reduced insulin sensitivity. 

### 4.3. Bile Acids

Paralleling previous studies [2,34], we observed a dramatic increase in several bile acids in response to the glucose challenge, including glycocholate, glycodeoxycholate, glycohyocholate, and taurocholate. In response to a meal, the gallbladder releases bile into the small intestine, stimulated by gastric filling and the intestinal hormone cholecystokinin (CCK). Post intestinal absorption and transport to the liver, it is estimated that 10–30% of bile acids reach systemic circulation [40]. Our results and others [2,34] have suggested that a bolus of glucose stimulates the release of bile acids in the gallbladder, supported by Liddle et al. finding that glucose ingesting stimulates CCK production [41]. Previous work has suggested a link between bile acid secretion and metabolism with obesity and IR [42]. At baseline, during the Fasted Visit, several primary and secondary bile acids were associated with obesity, including positive associations with chenodeoxycholate, hyodeoxycholate, and deoxycholate and inverse associations with glycocholate, glycohyocholate, taurocholate, and tauro-alpha/beta-muricholate. Furthermore, at t60 during the Fasted Visit, cholate and hyocholate were positively associated with IR in females (Table 2). Therefore, a blunted decrease of certain bile acids may be associated with insulin resistance and metabolic dysfunction. 

### 4.4. Conclustions and Future Directions

The metabolome response to an OGTT may be associated with IR in a sex-specific manner, due to the observed differences in insulin response to an OGTT in adolescent males and females. In healthy and metabolically unhealthy youth, insulin sensitivity decreases during puberty [43]. Furthermore, in a cohort of healthy children and adolescents, girls in late puberty (Tanner’s Stage 4 or 5) have higher insulin levels than boys [44]. The sexual dimorphism observed in late puberty is due, in part, to a higher growth hormone secretion in pubertal girls [45]. In our cohort, we observed higher levels of glucose t120, insulin t90, and insulin t120 in females, suggesting that females have a larger insulin response to the OGTT than males. Future analyses must be conducted to determine if the associations between the metabolome across visits and HOMA-IR in females is attributed to IR shifts during puberty or the onset of metabolic dysfunction and prediabetes. 

The metabolome was comprehensively profiled using a liquid chromatography/mass spectrometry-based platform, generating approximately 250 metabolites. Our study utilized a well-powered sample size, strongly complementing and elaborating on the only other study assessing the metabolome response to an OGTT in adolescents [11] by incorporating both a fasted and a random-fed visit in the OVOB participants. Our results emphasize the potential of analyzing the metabolome response in a random glucose challenge for the prediction of metabolic dysfunction, particularly in females. The results from this study emphasize that the switch from FA to glucose metabolism in response to a glucose challenge is associated with obesity and insulin resistance. Future work will collect plasma samples in response to a glucose challenge at more timepoints, such as 30 min, to assess more subtle changes in the metabolites, similarly to Zhao et al. [34]. 

Our study design presented two limitations regarding the Random-Fed Visit. Firstly, we only recruited OVOB participants for the visit, not allowing for a comparison between lean and OVOB. Secondly, we desired to simulate a random glucose challenge that is typically performed in practice for women being screened for gestational diabetes using 50-g of glucose. The differences in grams of glucose solution administered between the Fasted and Random-Fed Visit create challenges in the direct comparison of the metabolome response. Our priority was to replicate what is being practiced in the clinic. Future work should compare the metabolome response in different fed-states utilizing the same glucose load. A bioinformatic limitation in the study was the inability to map individual significant metabolites to biological pathways using MSEA, due to many metabolites with the pathways not being profiled in the untargeted metabolomics platform and the lack of HMDBs for metabolites that were significant. Future directions will incorporate a partial correlation-based approach [46] to assess alterations in the relationship of metabolites at the fasted random-fed visit and if subnetworks of metabolites are associated with insulin resistance cross-sectionally and longitudinally. 

Our results emphasize the utility of profiling multiple metabolic pathways outside glucose metabolism in understanding the associations between obesity, IR, and the response to a glucose challenge in adolescents. Classifying the metabolism of lipids, amino acids, and fatty acids, rather than solely glucose metabolism, deepens the understanding of the pathophysiology of insulin resistance in adolescents, with differences than adults due to pubertal development. Future work will test if the highlighted metabolic pathways complement or enhance the ability of glucose to predict the development of prediabetes during adolescence. 

## Figures and Tables

**Figure 1 nutrients-13-03365-f001:**
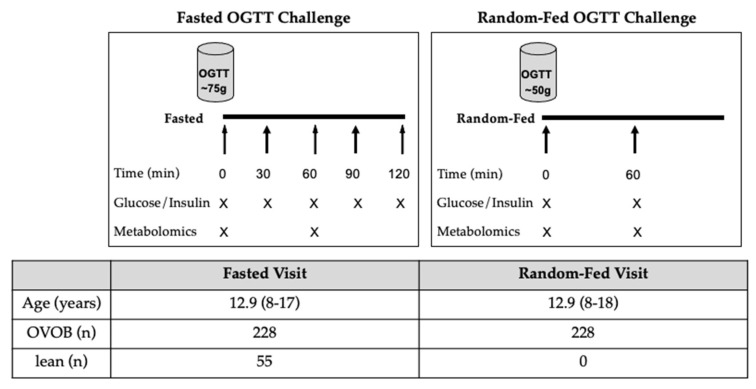
Study Design. Overweight and obese (OVOB) and lean participants were recruited prior to the first visit (Fasted Visit), where an oral glucose tolerance test (OGTT) (75 g) was administered in the fasted state. The OVOB participants returned approximately a week later for an OGTT (50 g) in a random fed state (Random-Fed Visit). Blood samples were collected before and during the OGTT and used for glucose, insulin, and untargeted metabolomics assays. Mean age and range of ages reported (years).

**Figure 2 nutrients-13-03365-f002:**
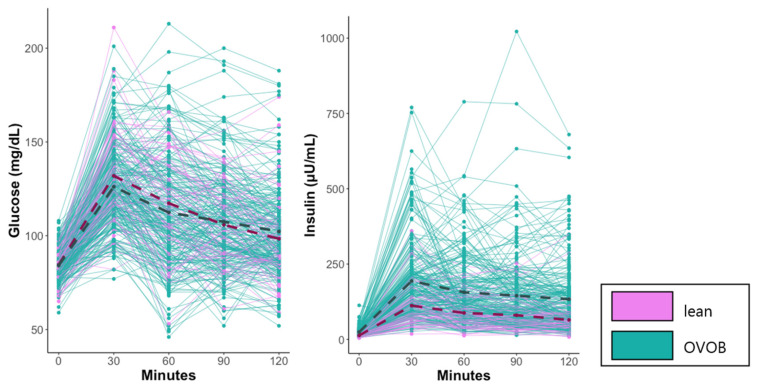
Kinetics of blood glucose and insulin response to the glucose test at the Fasted Visit. OVOB and lean participants arrived fasted prior to the consumption of the glucose challenge. Blood glucose and insulin were profiled before and during the OGTT. Mean values reported for lean (dark pink dash) and OVOB (dark slate gray dash). OVOB, overweight and obese.

**Figure 3 nutrients-13-03365-f003:**
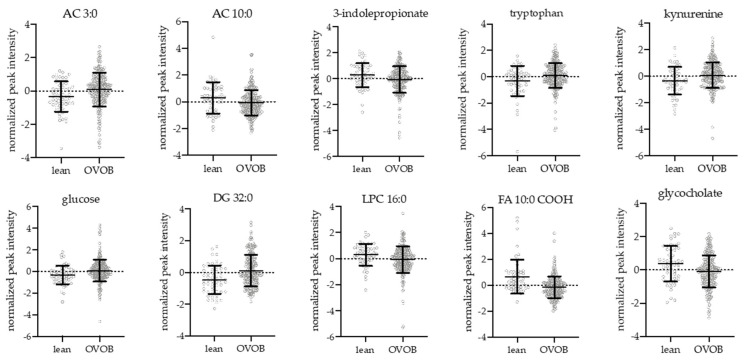
Differential metabolites between OVOB and lean participants during the Fasted OGTT Challenge. Linear regression analyses identified metabolites associated with the OVOB and lean groups at t0 (ß_group_), adjusting for sex and age (FDR < 0.1, 66 metabolites). Selected differential metabolites represent metabolic pathways associated with obesity at t0. Normalized peak intensities and standard deviations are reported.

**Figure 4 nutrients-13-03365-f004:**
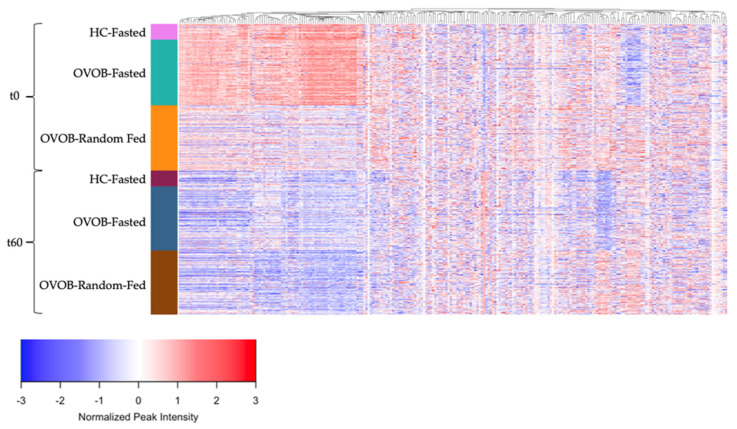
Dynamic response of the metabolome to an oral glucose tolerance test. Heatmap of normalized peak intensity for individual metabolites (*p* = 246) (mean 0, standard deviation 1). Metabolites are grouped by hierarchical clustering (columns). Subjects are ordered by group (OVOB and lean), time point after the OGTT (t0 and t60), and state (Fasted and Random-Fed) (rows). OVOB, overweight and obese.

**Figure 5 nutrients-13-03365-f005:**
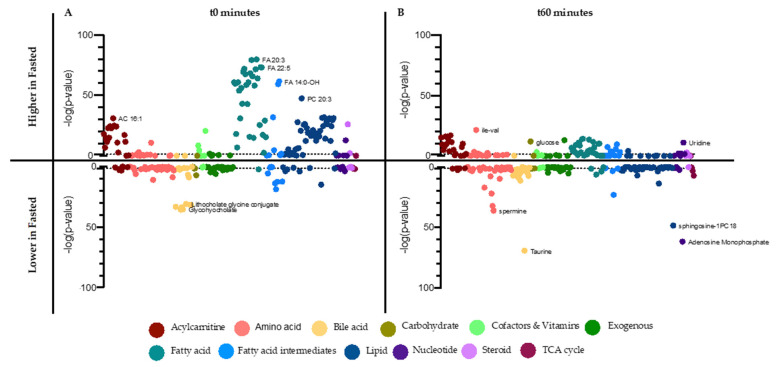
Differential metabolites between Fasted and Random-Fed in OVOB. Linear regression analyses identified metabolites associated with OVOB-Fasted and OVOB-Random-Fed (ßstate) at (**A**) t0 and (**B**) t60. On the top of the plots, metabolites are reported that are higher in OVOB-Fasted (ßstate < 0), with position indicated by −log10 (*p*-value). On the bottom of the plots, metabolites are reported that are lower in OVOB-Fasted (ßstate > 0), with position indicated by −log10 (*p*-value). Metabolites are listed in the same order in 5A and 5B. Colors indicate metabolite class. Horizontal lines (dotted) signify FDR = 0.1. Several distinguishing metabolite names are listed. Age and sex are included in the model.

**Table 1 nutrients-13-03365-t001:** Characteristics of study participants at the Fasted Visit, stratified by weight group. Significance denoted with unadjusted *p*-value < 0.05 (bolded).

**Categorical Variables**	**OVOB**	**Lean**	** *p* ** **-Value ^1^**
	** *n* ** **(%)**	** *n* ** **(%)**	
**sex**			
Male	97 (43%)	26 (47%)	0.5254
Female	131 (57%)	29 (53%)	
race			
Asian/Pacific Islander	4 (2%)	4 (7%)	0.1745
African American/Black	59 (26%)	12 (22%)	
White	135 (59%)	32 (58%)	
more than one race	19 (8%)	6 (11%)	
did not wish to report	11 (5%)	1 (2%)	
ethnicity			
Hispanic	18 (8%)	5 (9%)	0.7707
non-Hispanic	210 (92%)	50 (91%)	
abnormal 2-hr plasma glucose (≥140 mg/dL)			
Yes	16 (7%)	3 (5%)	0.6776
No	212 (93%)	52 (95%)	
abnormal fasting plasma glucose (≥100 mg/dL)			
Yes	8 (4%)	1 (2%)	0.5213
No	220 (96%)	54 (98%)	
ADA prediabetes (FPG≥ 100 mg/dL or 2-hr PG ≥ 140 mg/dL or HbA1c ≥ 5.7%)			
Yes	35 (15%)	6 (12%)	0.4009
No	193 (85%)	49 (88%)	
**Continuous Variables**	**OVOB**	**Lean**	** *p* ** **-Value ^2^**
	**Mean (SD)**	**Mean (SD)**	
age (years)	12.9 (2.5)	13.0 (2.6)	0.7301
BMI percentile	95 (4)	59 (27)	**7.52** ^−14^
HOMA-IR	5.13 (2.99)	2.80 (1.21) ^3^	**4.53** ^−8^
HbA1c	5.2 (0.3)	5.1 (0.3) ^3^	0.2082
fast time (hours)	14.0 (1.3)	14.1 (1.4)	0.7606
fasting OGTT response			
glucose (t0) (mg/dL)	84 (8)	85 (8)	0.7475
glucose (t30) (mg/dL)	126 (22)	132 (25)	0.1119
glucose (t60) (mg/dL)	112 (29)	117 (26)	0.2182
glucose (t90) (mg/dL)	107 (26)	106 (21)	0.6263
glucose (t120) (mg/dL)	102 (24)	98 (22)	0.2562
insulin (t0) (µU/mL)	24 (14)	13 (5) ^3^	**1.55 × 10^−8^**
insulin (t30) (µU/mL)	194 (132)	112 (72)	**4.09 × 10^−9^**
insulin (t60) (µU/mL)	156 (117)	87 (53)	**7.04 × 10^−10^**
insulin (t90) (µU/mL)	145 (126)	80 (56)	**2.69 × 10^−8^**
insulin (t120) (µU/mL)	133 (117)	65 (57)	**2.42 × 10^−9^**
AUC glucose	3121 (2112)	3294 (1711)	0.5228
AUC insulin	17,223 (11,206)	9648 (5468)	**1.33 × 10^−11^**

^1^ Represents Pearson’s chi-square test for categorical variables. ^2^ Represents unpaired *t*-test for continuous variables. ^3^
*n* = 54.

**Table 2 nutrients-13-03365-t002:** Metabolites associated with HOMA-IR in females with overweight and obesity at the Fasted and Random-Fed Visit. Beta coefficients and standard errors from linear regression models are reported, adjusting for age at the Fasted Visit (FDR < 0.1).

Metabolite	Pathway	Fasted t0	Fasted t60	Fasted Fold Change	Random-Fed t0	Random-Fed t60
AC 12:0	acylcarnitine			2.0 ± 0.3		
AC 12:1	acylcarnitine			2.7 ± 0.5		
AC 14:0	acylcarnitine			2.3 ± 0.8		
AC 16:0	acylcarnitine			2.6 ± 0.6		
AC 16:1	acylcarnitine			1.3 ± 0.7		
AC 18:0	acylcarnitine			1.6 ± 0.2		
AC 5:0-OH	acylcarnitine		0.9 ± 0.3			
AC 5:1	acylcarnitine			1.5 ± 1.6		
AC 6:0	acylcarnitine			1.7 ± 0.6		
gamma-glutamyltyrosine	amino acid	0.8 ± 0.2	0.9 ± 0.3		0.9 ± 0.3	
Glu-Phe	amino acid		0.9 ± 0.2			
glutamate	amino acid		0.7 ± 0.3			0.9 ± 0.3
indole-3-methyl acetate	amino acid		0.7 ± 0.3			
L-gamma-glutamylisoleucine	amino acid		0.8 ± 0.3			
Leu-Ile	amino acid			0.7 ± 0.5		0.9 ± 0.2
leucine+isoleucine	amino acid		0.7 ± 0.3			
N-acetylphenylalanine	amino acid		0.7 ± 0.3			
Phe-Phe	amino acid			−0.7 ± 0.3		
Phe-Trp	amino acid		0.7 ± 0.3			
pipecolate	amino acid			−0.6 ± 1.1		
proline	amino acid		0.7 ± 0.3			
cholate	bile acid		0.8 ± 0.3			
hyocholate	bile acid		0.8 ± 0.3			
indole-3-lactate	carbohydrate		0.7 ± 0.3			
caffeine	exogenous			1.7 ± 0.6		
FA 18:4	fatty acid		0.8 ± 0.3			
FA 20:3	fatty acid			1.1 ± 0.3		
FA 22:1	fatty acid			1.9 ± 0.5		
3-hydroxyphenyl-valerate	fatty acid intermediate			0.9 ± 0.6		
DG 32:0	lipid	1.2 ± 0.2	1.2 ± 0.2	2.2 ± 0.4	1.1 ± 0.2	1.0 ± 0.2
DG 32:1	lipid	1.0 ± 0.2	1.1 ± 0.2		0.9 ± 0.2	0.8 ± 0.2
DG 34:1	lipid		1.0 ± 0.2		0.9 ± 0.2	0.9 ± 0.2
DG 34:2	lipid	1.1 ± 0.2	1.0 ± 0.2		0.9 ± 0.2	0.8 ± 0.2
DG 36:2	lipid		0.7 ± 0.2			
DG 36:3	lipid		0.7 ± 0.3			
MG 14:0	lipid					1.0 ± 0.2
MG 16:0	lipid			1.2 ± 0.3		
MG 18:1	lipid	0.8 ± 0.2	0.9 ± 0.2			0.7 ± 0.3
LPC 16:0	lipid			1.1 ± 1.2		
LPC 18:2	lipid			0.8 ± 1.7		
PC 32:1	lipid		0.8 ± 0.2	1.0 ± 0.4		
PC 34:3	lipid		0.7 ± 0.3			
PC 34:4	lipid		0.7 ± 0.3			
N2,N2-dimethylguanosine	nucleotide		0.7 ± 0.3			
urate	nucleotide	0.8 ± 0.2	1.0 ± 0.2		1.0 ± 0.2	0.9 ± 0.2

## Data Availability

Data described in the manuscript, code book, and analytic code will be made available upon request. Metabolomics data is available at the National Metabolomics Data Repository (metabolomicsworkbench.org/data/index. php).

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
