# Peer review of "Comparing the Fasting and Random-Fed Metabolome Response to an Oral Glucose Tolerance Test in Children and Adolescents: Implications of Sex, Obesity, and Insulin Resistance"

_nutrients, 2021, doi:10.3390/nu13103365_

Round 1
Reviewer 1 Report
In this study, metabolic changes of lean and overweight/obese adolescents were assessed using oral glucose tolerance test (OGTT), and metalomic results associated to the test were evaluated. Interestingly, obesity was associated with a decline response of fatty acid oxidation intermediates. In girls, fasted and non-fasted OGTT results were associated with HOMA-IR.
In regards to some questions, is it possible for the authors to explain why the fasting OGTT was performed up to 120 min and used 75 g of glucose while non-fasted OGTT was performed up to 60 min and used 50 g of glucose? Wouldn’t it be more appropriate for comparison purposes if the OGTT test was similar between fasted and non-fasted?
Secondly, was there any sex differences in the metabolimics results? It would be interesting besides segregating the groups between lean vs overweight/obese, but also female lean vs female overweight/obese, and male lean vs male overweight/obese and combinations for better sex differences responses.
What is the clinical relevance of this study? Could you please comment on the conclusion section?
Author Response
Reviewer 1. Comments and Suggestions for Authors
In this study, metabolic changes of lean and overweight/obese adolescents were assessed using oral glucose tolerance test (OGTT), and metabolomic results associated to the test were evaluated. Interestingly, obesity was associated with a decline response of fatty acid oxidation intermediates. In girls, fasted and non-fasted OGTT results were associated with HOMA-IR.
Comment 1. In regards to some questions, is it possible for the authors to explain why the fasting OGTT was performed up to 120 min and used 75 g of glucose while non-fasted OGTT was performed up to 60 min and used 50 g of glucose? Wouldn’t it be more appropriate for comparison purposes if the OGTT test was similar between fasted and non-fasted?
Response. For formal classification of diabetes status, we conducted OGTTs that used 75 g of glucose performed up to 120 minutes because that is the gold standard protocol OGTT assessment. For the non-fasting OGTT, we simulated an oral non-fasting OGTT that is typically performed in practice for women being screened for gestational diabetes. While there are differences between the fasting and non-fasting OGTT within the study, including the grams of glucose solution administered, we desired to replicate what is being practice in the clinic. In response to this comment, we’ve included stronger language describing our study design aim in the methods (lines 113-116) and more acknowledgment of the limitations that this design poses (lines 495-501).
Comment 2. Secondly, was there any sex differences in the metabolomics results? It would be interesting besides segregating the groups between lean vs overweight/obese, but also female lean vs female overweight/obese, and male lean vs male overweight/obese and combinations for better sex differences responses.
Response. In alignment with our work and others, we do observe sex differences in the metabolome at t0 and t60 minutes post-OGTT challenge (Table S3 and Lines 282-289). In response, it would be ideal to stratify by males and females when assessing the relationship between obesity group and metabolites (Table S2). However, the number of subjects within each group becomes very small in the leans: lean males (n=26), lean females (n=29), OVOB males (n=97), and OVOB females (n=131). Therefore, our cohort did not provide enough power to conduct a sex-stratified analysis determining the influence of BMI group on metabolites.
In response to your comment, we did check for sex-differences in the metabolome response to the OGTT. We conducted a paired t-test of each metabolite response for males at the Fasted Visit (t0 vs. t60), males at the Non-Fasted Visit (t0 vs. t60), females at the Fasted Visit (t0 vs. t60), and females at the Non-fasted Visit (t0 vs. t60). Next, we correlated the fold change for all metabolites (log2(t60/t0)) between males and females at the Fasted Visit and the Non-fasted visit, observed little sex-differences in the metabolite repones to the OGTT. At the Fasted Visit, the correlation of the metabolite fold changes was 0.98 (r2) between males and females. At the Non-Fasted Visit, the correlation of the metabolite fold changes was 0.88 (r2) between males and females. These results suggest similar response of the metabolome by sex. The metabolites with the largest sex-difference in their fold changes were the fatty acids, which we’ve discussed were elevated in females at fasting (Lines 285).
Comment 3. What is the clinical relevance of this study? Could you please comment on the conclusion section?
Response. Our group desires to identify metabolite predictors of prediabetes in adolescents using the metabolome response to a glucose challenge. The proposed manuscript highlights lipid, amino acid, and fatty acid pathways that are altered in response to a OGTT and associated with obesity and insulin resistance. This cross-sectional analysis was necessary to provide the framework for longitudinal prediction analyses that will leverage highlighted metabolites. These metabolic pathways may provide stronger predictive ability than profiling solely glucose metabolism for the prediction of prediabetes development in adolescents (Lines 508-515).
Reviewer 2 Report
The first part of the study is an excellent. The main objective was to assess the metabolome response to an OGTT performed in fasting state in young subjects under the age of 17 years and compare the results of 228 overweight/ obesity participants with that in 55 lean subjects. OGTT was performed, with the glucose load of 1.75 g/kg body weight (max. 75 g), after a mean 14 h fasting time (range 9-19 h). Plasma glucose and insulin were measured at time 0’, 30’, 60’, 90’, 120’ of OGTT. Untargeted metabolomics profiled 246 metabolites in plasma at time 0’ and at 60’ of OGTT. Obtained results were reffered to BMI, age, sex, and insulin-resistance (IR) assessed by HOMA-IR. The results suggest that obesity and IR starting at a young age influence the switch from fatty acid to glucose oxidation in response to oral glucose load, as was evidenced by reductions in fatty acid oxidation intermediates, including acylcarnitines.
Small remarks concerning this part:
- The title is a little misleading; the study was conducted in subjects aged 8-17 years old, so it comprised also children, not only adolescents.
- The sentence concerning OGTT (Introduction, line 43-45) should be corrected a little, as IGT is the same as prediabetes. Additionally, if OGTT is used classically for diagnostic purposes, insulin response to ingestion of 75 glucose load is not measured (the same lines).
- Abbrevations used in the text should be elucidated when they occur for the first time (e.g – BCAAs (line 57), LPC (line 265), DGs, PCs, SMs (lines 266-267).
- The fact that prediabetes (ADA criteria 2020) was diagnosed in similar percentage of owerweight/obese and lean subjects (15% and 12%, respectively) is surprising and needs an explanation or at least a comment.
The second part of the study concerning non-fasting metabolome response to OGTT rises serius doubts. In overweight/obese participants the second OGTT was performed in non-fasted state and the metabolome results were compared with the fasted- state OGTT results. The main objection regards the methodology of performing OGTT in non-fasted state.
Major objections concerning this part:
- Te second OGTT regarded as a non-fasting test, in fact was performed in random glucose state. The fasting times before this OGTT veried from 5 minutes to 14 hours (!), so in some persons the investigation was conducted at fasting state, exactly as at the first OGTT. In others it was performed shortly after the meal.
- Glucose load at the second OGTT was 50 g in all participants. In the first test the load was 1.75g/kg body weight. According to this fact, some subjects (weight approx. 28-29 kg) were given the same glucose load of 50 g in both tests, while the others (weight >42 kg) were loaded with different amount of glucose; 75 g in the first, and 50 g in the second test.
- It is really difficult to compare glucose, insulin and other metabolities response in such great varibility of conditions of performing both OGTTs, which were applied in different subjects from the same group.
Author Response
Reviewer 2. Comments and Suggestions for Authors
The first part of the study is an excellent. The main objective was to assess the metabolome response to an OGTT performed in fasting state in young subjects under the age of 17 years and compare the results of 228 overweight/ obesity participants with that in 55 lean subjects. OGTT was performed, with the glucose load of 1.75 g/kg body weight (max. 75 g), after a mean 14 h fasting time (range 9-19 h). Plasma glucose and insulin were measured at time 0’, 30’, 60’, 90’, 120’ of OGTT. Untargeted metabolomics profiled 246 metabolites in plasma at time 0’ and at 60’ of OGTT. Obtained results were reffered to BMI, age, sex, and insulin-resistance (IR) assessed by HOMA-IR. The results suggest that obesity and IR starting at a young age influence the switch from fatty acid to glucose oxidation in response to oral glucose load, as was evidenced by reductions in fatty acid oxidation intermediates, including acylcarnitines.
Small remarks concerning this part:
Comment 1. The title is a little misleading; the study was conducted in subjects aged 8-17 years old, so it comprised also children, not only adolescents.
Response. We changed the title to “Comparing the fasting and random-fed metabolome response to an oral glucose tolerance test in children and adolescents: Implications of sex, obesity, and insulin resistance.”
Comment 2. The sentence concerning OGTT (Introduction, line 43-45) should be corrected a little, as IGT is the same as prediabetes. Additionally, if OGTT is used classically for diagnostic purposes, insulin response to ingestion of 75 glucose load is not measured (the same lines).
Response. We made the following clarifications in lines 43-45: “Classically, an oral glucose tolerance test (OGTT) diagnoses impaired glucose tolerance (IGT)/prediabetes and T2D, by measuring the acute trajectory of glucose in response to ingestion a 75-gram glucose solution.”
Comment 3. Abbrevations used in the text should be elucidated when they occur for the first time (e.g – BCAAs (line 57), LPC (line 265), DGs, PCs, SMs (lines 266-267).
Response. The following abbreviations were defined: branched chain amino acids (BCAA, line 57), lysophosphocholine (LPC, line 269), diglyceride (DG, line 271), phosphocholine (PC, line 271), and sphingomyelin (SM, line 271).
Comment 4. The fact that prediabetes (ADA criteria 2020) was diagnosed in similar percentage of owerweight/obese and lean subjects (15% and 12%, respectively) is surprising and needs an explanation or at least a comment.
Response. In response, we further explored subject differences between the overweight/obese and lean subjects with prediabetes. Three of the six lean participants that classified as prediabetic had a BMI percentile of 84%, potentially explaining why group trends were not observed (Lines 218-220). The other three individuals were on the border of prediabetic, with an A1c of 5.7%, even though their BMI percentile was not approaching overweight or obese. Interestingly, these three individuals were pubertal, potentially suggesting alterations in their insulin regulation due to pubertal development.
The second part of the study concerning non-fasting metabolome response to OGTT rises serious doubts. In overweight/obese participants the second OGTT was performed in non-fasted state and the metabolome results were compared with the fasted- state OGTT results. The main objection regards the methodology of performing OGTT in non-fasted state.
Major objections concerning this part:
Comment 5. The second OGTT regarded as a non-fasting test, in fact was performed in random glucose state. The fasting times before this OGTT veried from 5 minutes to 14 hours (!), so in some persons the investigation was conducted at fasting state, exactly as at the first OGTT. In others it was performed shortly after the meal.
Response. As Supplemental Figure 1c reports, the fasting time prior to the OGTT varied greatly. The objective of conducting a Non-Fasting OGTT was to assess the utility of a glucose challenge for the prediction of prediabetes in children and adolescents at any random fed/fast state. Therefore, the utility of this study design is to determine if children and adolescents can be brought into the clinic at a random state for a OGTT test for prediction. We recognize that calling the visit “Non-Fasted” wasn’t the proper classification. Therefore, throughout the manuscript, we changed the text “Non-Fasted” to “Random-Fed State.”
Comment 6. Glucose load at the second OGTT was 50 g in all participants. In the first test the load was 1.75g/kg body weight. According to this fact, some subjects (weight approx. 28-29 kg) were given the same glucose load of 50 g in both tests, while the others (weight >42 kg) were loaded with different amount of glucose; 75 g in the first, and 50 g in the second test.
Response. Individuals that received both the Fasted and Random-Fed State OGTT were overweight and obese (OVOB). The Fasted OGTT was indeed adjusted for weight (1.75g/kg) in accordance with the American Diabetes Association protocol for children. Supplemental Figure 1b highlights there were individuals who received less than 75g of glucose at the Fasted OGTT. Fifteen OVOB participants received less than 75g, with one receiving <60g, twelve receiving between 60g and 70g, and two receiving >70g. Therefore, only one OVOB individual received a load comparable (<60g) to the second OGTT (Random-Fed State).
Comment 7. It is really difficult to compare glucose, insulin and other metabolities response in such great varibility of conditions of performing both OGTTs, which were applied in different subjects from the same group.
Response. Yes, we acknowledge that our study design did not control for glucose administered, but rather was created to replicate the gold standard OGTT for diabetes diagnosis and the OGTT typically performed in practice for women being screened for gestational diabetes. In response to this comment, we’ve included stronger language describing our study design aim in the methods (lines 113-116) and more acknowledgment of the limitations that this design poses (lines 495-501).
Round 2
Reviewer 2 Report
Hoever some doubts concerning methodology are still present, but performed corrections and explanation added to the section "limitation of the study" are sufficientt and satisfying
Author Response
We will consider your suggestions for future research designs. Thank you for your comments during our revision.